# Understanding the Role of Connectivity Dynamics of Resting-State Functional MRI in the Diagnosis of Autism Spectrum Disorder: A Comprehensive Study

**DOI:** 10.3390/bioengineering10010056

**Published:** 2023-01-02

**Authors:** Yaser ElNakieb, Mohamed T. Ali, Ahmed Elnakib, Ahmed Shalaby, Ali Mahmoud, Ahmed Soliman, Gregory Neal Barnes, Ayman El-Baz

**Affiliations:** 1Bioengineering Department, University of Louisville, Louisville, KY 40292, USA; 2Department of Neurology, Pediatric Research Institute, University of Louisville, Louisville, KY 40202, USA

**Keywords:** MRI, rs-fMRI, CAD, ML, autism, ASD

## Abstract

In addition to the standard observational assessment for autism spectrum disorder (ASD), recent advancements in neuroimaging and machine learning (ML) suggest a rapid and objective alternative using brain imaging. This work presents a pipelined framework, using functional magnetic resonance imaging (fMRI) that allows not only an accurate ASD diagnosis but also the identification of the brain regions contributing to the diagnosis decision. The proposed framework includes several processing stages: preprocessing, brain parcellation, feature representation, feature selection, and ML classification. For feature representation, the proposed framework uses both a conventional feature representation and a novel dynamic connectivity representation to assist in the accurate classification of an autistic individual. Based on a large publicly available dataset, this extensive research highlights different decisions along the proposed pipeline and their impact on diagnostic accuracy. A large publicly available dataset of 884 subjects from the Autism Brain Imaging Data Exchange I (ABIDE-I) initiative is used to validate our proposed framework, achieving a global balanced accuracy of 98.8% with five-fold cross-validation and proving the potential of the proposed feature representation. As a result of this comprehensive study, we achieve state-of-the-art accuracy, confirming the benefits of the proposed feature representation and feature engineering in extracting useful information as well as the potential benefits of utilizing ML and neuroimaging in the diagnosis and understanding of autism.

## 1. Introduction

Autism spectrum disorder (ASD) is a neurodevelopmental disorder characterized by three primary characteristics: functioning difficulties with social interaction, communication barriers, and behavioral restrictions and repetitive patterns [1,2,3,4]. Despite the lack of a comprehensive understanding of ASD causes, numerous hypotheses and theories have been proposed concerning the etiology of its underlying mechanism. These hypotheses and theories suggest that genes and environmental factors play a significant role in determining ASD severity. Anatomical abnormalities of the brain [5], functioning of the brain during rest or while performing different tasks [6], or abnormal connectivity of the white matter [7,8] are hypothesized to be responsible for ASD symptoms. Several magnetic resonance imaging (MRI)-based imaging methods have been utilized to study a variety of abnormalities correlated with ASD, including: (i) structural magnetic resonance imaging (sMRI) for anatomical anomalies; (ii) functional magnetic resonance imaging (fMRI), either at rest or while performing a task, for abnormalities in brain activity; (iii) diffusion tensor imaging (DTI) for abnormalities in connectivity. In this study, we utilize fMRI to provide an understanding of assessing autism.

There are two major types of experiments used to analyze the fMRI modality in order to study functional activation anomalies in the brain: (i) resting state fMRI (rs-fMRI) and (ii) task-based fMRI [9]. In task-based functional MRI, the subject performs a particular task, such as: (1) figures [10], (2) facial expressions [11], (3) rewards [12], or (4) other specific tasks, and then, the brain activity is analyzed. In rs-fMRI, brain activity is captured, in terms of blood-oxygen-level-dependent (BOLD) signals, while the subject is at rest. Since connectivity patterns reveal the resting state, they have proven to be beneficial in diagnosing mental disorders, such as schizophrenia [13] and Alzheimer’s disease [14]. For instance, a high degree of accuracy was achieved in diagnosing schizophrenia using deep neural networks [13]. In addition, another deep-learning approach used rs-fMRI and ResNet to achieve an effective multi-class classification of Alzheimer’s disease [14]. There is evidence that the functional connectivity between major networks, as well as the functional connectivity within individual networks, is altered in ASD, as demonstrated by numerous studies [15,16]. Based on a multivariate autoregressive model, an algorithm based on a machine learning algorithm for analyzing functional connectivity in ASD subjects was used in a previous study [17]. In this study, it was found that people with ASDs have reduced functional connectivity, which supports the hypothesis that autism is characterized by a deficit in functional connectivity, or ’underconnectivity’. In accordance with the underconnectivity theory [18], cognitive disorders manifest themselves in reduced synchronized brain activity during integrative processing tasks, such as the synthesis of a sentence from a set of words. There is evidence in previous literature [17,19] for the hypothesis of underconnectivity. Similarly, there was lower functional connectivity in the superior parietal and visuospatial regions of ASD when compared to TD [10]. In another study by Tyszka et al. [20], there was reduced connectivity between the temporal and frontal cortex, but no global abnormalities were found. In a study by Plitt et al. [21], abnormalities were reported in functional networks, which were more evident in networks related to social information processing.

Nevertheless, brain networks associated with ASDs exhibit both under-connectivity and over-connectivity [21,22]. Unlike previous studies, ASD has not only reported results that indicated brain underconnectivity but also increased brain connectivity in some areas when compared to healthy controls [22]. In the latter study, functional connectivity patterns were analyzed, specifically interhemispheric connectivity analysis, and it was found that ASD subjects had both under- and over-connectivity. Furthermore, the presence of altered connectivity was verified in Di Martino et al. work [23]. The subjects with ASD displayed both under- and over-connectivity between different brain regions. Furthermore, a study by Supekar et al. [24] identified hyperconnectivity in severely socially challenged autistic children.

Providing the appropriate reduction technique is another major component of our proposed work. Bellman [25] introduced the term ‘curse of dimensionality’ to describe the problem of exponential complexity resulting from the addition of new dimensions to feature space, commonly defined as the p>>n problem. MRI imaging research and medical data are commonly affected by this phenomenon, resulting in over-fitting. There are several ways to reduce the number of features, such as principal component analysis, linear discriminant analysis, or autoencoders [26,27,28]. Due to the fact that these methods do not preserve the semantics of the original feature space, it is typically difficult to identify what clinical findings underlie such classification results. This, in turn, makes them less useful for actual use by physicians to provide them with more information or to better understand the pathological abnormalities underlying each autistic brain and, thus, are less attractive and less practical.

Due to autism’s heterogeneous nature, brain connectivity can vary widely among individuals with the disorder, making classification of the condition difficult. The design of a less-sensitive functional connectivity feature that is less affected by age, sex, and designs of the resting-state scan and studying its correlation with autism is a hot research area [29], emerging into two popular brain connectivity representations: the conventional, more popular, static functional connectivity (FC), and the newer dynamic functional connectivity (dFC) representations. Those functional connectivity metrics are believed to capture different internal states of the brain while at rest [30]. The static FC is a matrix obtained by calculating the Pearson cross-correlation coefficient of the BOLD signals across pairs of pre-defined brain areas. Following statistical analysis and depending on whether one examines local or global networks, there is evidence of both under- and over-connectivity in autism in the majority of the literature [29]. As most of the literature implies temporally stationary functional networks in resting state, dynamic or time-varying changes to functional connectivity that occur during brain scanning are not sufficiently considered by static FC. Newer studies suggest periodically changing spatial patterns of functional networks. To capture those dynamic connectivity patterns, multiple computational strategies were used to find dFC that consider such temporal fluctuations of functional connectivity [31]. dFC analyses allow identifying not only common brain states but also transitions between them. The most commonly used approach for dFC computation is sliding window techniques [32], while other approaches include clustering methods, dynamic connectivity regression, time–frequency analysis, wavelet transforms [33], dynamic connectivity detection, and time series models [31,34].

The main limitation of the previous works is that each only gives a limited perspective on the bolts of an efficient CAD system. As the characteristics of autism vary from individual to individual in terms of symptoms and severity, a more personalized approach to predicting and analyzing the behavior and functional capabilities of each autistic subject has become increasingly necessary. Consequently, we can create a treatment plan that is tailored specifically to the needs of each autistic individual. This work uses both a static FC matrix, as well as a novel dFC approach, to investigate the role of rs-fMRI in the diagnosis of autism in a large cohort of 884 subjects, testing different processing and machine learning pipelines. It allows not only an accurate diagnosis but also the identification of the brain regions contributing to this decision. More importantly, it quantifies the effect of different choices along the neuroimaging machine learning pipeline on diagnostic accuracy, including the use of dynamic over static connectivity, which is important for future research. This work presents a unified framework that addresses the following:The impact of using different atlases, including the automated anatomical labeling (AAL) and Talaraich and Tournoux (TT) atlases.The effect of using different preprocessing strategies.The effect of using our novel dFC, in comparison to using conventional static FC.The dimensionality reduction problem, using two-stage feature selection, with four types of kernels.The role of the classification strategy, investigating six different classifiers.The ability to highlight the importance of each of the previous choices on the overall performance.

The remainder of this paper is organized as follows: Section 2 describes the entire methodology of feature representation, preprocessing, and diagnostic classification. We present the various results on the ABIDE-I rs-fMRI in Section 3. Section 4 provides an overall discussion of the presented work. Finally, the conclusions are presented in Section 5.

## 2. Materials and Methods

### 2.1. Dataset

In this work, the Autism Brain Imaging Data Exchange I (ABIDE I) dataset is used [35], which is a publicly available repository collected from 17 different sites, with rs-fMRI, sMRI imaging modalities (http://fcon_1000.projects.nitrc.org/indi/abide/abide_I.html, accessed on 1 July 2022). The database includes 884 subjects with rs-fMRI data from the 17 sites of the dataset. All of the 884 rs-fMRI data, are used in our experiments. Table 1 provides the summary of the 884 subjects’ demographics: age in years, label, full-scale IQ, and gender. Appendix A provides the full demographics of the subjects: subject IDs, along with age, label, IQ, and gender, while the scanning parameters of the functional MRI data for each of the 17 sites are available on the dataset website http://fcon_1000.projects.nitrc.org/indi/abide/abide_I.html, accessed on 1 July 2022. The parameters include the type of scanner, the scanning protocol, repetition time (TR), echo time (TE), flipping angle, and experiment duration for resting state fMRI.

### 2.2. Proposed Framework

Figure 1 illustrates the proposed framework, which starts with data preprocessing, brain parcellation to local areas, feature engineering, feature selection, and machine learning toward final diagnosis. Our proposed quantitative research model includes the identification of the effects of feature representation, prepossessing strategy, and atlas on diagnostic accuracy.

### 2.3. Preprocessing

Functional MRI data are preprocessed according to the standard configurable pipeline for the analysis of connectomes (C-PAC) [36]. Preprocessing pipeline includes performing slice timing correction, motion realignment, skull stripping, and intensity normalization in this order. Twenty-four motion parameters were used as regressors to overcome the subject’s motion-related confounding variables. Additionally, five more from white matter and CSF mean levels were regressed to eliminate their effects. Then, one of four different preprocessing strategies is applied for filtration and signal correction. The main differences between those strategies are whether global signal correction (inclusion of global mean signal in nuisance regression) is made and whether filtration (band-pass filtering after global signal correction) is performed. Those four strategies are denoted as: filt_global, filt_noglobal, nofilt_global, and filt_noglobal. Lastly, structural MRI data, after skull stripping, segmentation, normalization, and registration to the standard MNI-152 space, is used for registration of the corresponding fMRI volumes to the same space after those preprocessing steps. For each of the four strategies, we will have different outputs.

Different brain regions are identified on the registered data using a standard atlas parcellation. The use of a standard atlas is to localize the fMRI features (time-series BOLD signals) to different brain regions, as defined by each atlas. Two atlases are investigated in this work: automated anatomical labeling (AAL) atlas, defining 116 brain regions, and Talaraich and Tournoux (TT) atlas, defining 97 different regions using Brodmann area labeling. In this work, all brain regions defined by each of the two used brain atlases are analyzed. For each atlas, mean BOLD signals for each parcellated brain region are extracted.

After preprocessing, each subject is now represented by an average BOLD signal (time series, length varying by site based on total scanning time) for each brain region, and we have eight datasets, one for each (strategy/atlas). The next step is to convert it to a more meaningful representation that captures functional connectivity and is less dependent on the subjects’ differences.

### 2.4. Feature Representation

For the purpose of studying the coherence between different areas of the brain, two different feature representations were used to examine functional connectivity. First, the most famous and commonly used static FC matrix, where the correlation between the full duration of the activation courses is used as a measure of functional connectivity. The reason behind this selection is that it captures the intrinsic functional network of the brain.

The static functional connectivity matrix FC is constructed as the Pearson correlation coefficient (ρ) between each pair of the (A=116|97 ) areas in the atlas. The effective output feature size is (A×(A−1)÷2), because of symmetry, resulting in a total of 884×6670 for AAL, or 884×4656 for TT, feature matrices. Figure 2 illustrates the pipeline of this adopted feature representation.

The second feature representation introduced in this work is an enhanced version of dynamic functional connectivity, dFC, where temporal dynamics are considered in the correlation calculation. The calculation starts by multiplying a shorter Gaussian sliding window with width *w* with each time signal, to calculate pair correlation, with an overlapping step size *s*. Following previous literature [34], a Gaussian window of size w=21TR, σ=3TR, and a step size of s=1 are used, where TR is the fMRI repetition time, typically 1500–2000 ms depending on the scanning site. Hence, for each pair of brain regions of length *L*, *M* correlations are calculated, where M=||(L−w)/s||. Following the hypotheses of under-connectivity and over-connectivity differences between the functional activation of an autistic and typically developed brain, a quantification using both the percentage of strong correlations $n_{st}$ and the percentage of weak/no correlation $n_{wk}$ are used to represent over- and under-connectivity between each brain area, where $n_{st}=\frac{1}{M}\sum_{M}^{}1:\;if\;\rho_{ij}\geq 0.8$, and $n_{wk}=\frac{1}{M}\sum_{M}^{}1:\;if\;\rho_{ij}\leq 0.25$. This creates two metrics for each pair of regions identifying the existence of under- and over-connectivities, yielding double size dFC feature matrices in comparison to the FC representation.  Figure 3 illustrates the pipeline of the second proposed feature representation in this study. At this step, we have 16 different inputs, one for each (strategy/atlas/feature representation) combination.

### 2.5. Feature Selection

Since the feature space for both representations is large, there is an urge for the employment of a feature reduction technique to address the curse of the dimensionality problem. Since the employed feature reduction shall preserve the semantics of original features, many famous techniques such as principal component analysis (PCA), factor analysis (FA), or linear discriminant analysis (LDA) may not be suitable, as it creates a newly transformed feature space that makes it hard to extrapolate the meaning of any following clinical classification, or underlying functional abnormalities of autistic individuals. Two cascaded feature selection stages are used to select the best subset of features: a simple univariate selector followed by a more sophisticated recursive feature elimination (RFE) with cross-validation (RFE-CV).

The univariate selector is a fast method that aims to initially reduce the feature space to make the second stage computationally efficient. It computes the ANOVA F-value score for each feature to select the top proportion accordingly. Following this, an RFE eliminates weak features until a specified number of features is reached by fitting a kernel classifier and removing the weakest features. By eliminating dependencies and collinearities, RFE attempts to achieve a better understanding of the model. Cross-validation aims to score the strength of features on the test subset, other than the training folds data used in training the kernel, to mitigate over-fitting. The step of the eliminated features, scoring metric, kernel type, and the number of folds are all parameters of choice. Here, a step size of 2 features (for faster elimination, yet small enough for a good performance), k = 5 folds (common 5-fold cross-validation), and balanced accuracy were chosen. Four different classifiers were tested as a feature selection kernel, namely linear support vector machine (lsvm), logistic regression (lr), random forest (rf), and light gradient boost machine (lgbm) [37,38], to investigate different feature-relationships. The two-stage feature selection is applied to each dataset/feature representation to find the best *n* features that provided the best cross-validated score. Next, the performance of the sixteen models was evaluated on a set of machine learning classifiers to select which model to use for further processing. The code for all of the previous steps, starting from data download, processing, and feature calculations, is publicly available on reference [39].

### 2.6. Machine Learning

In order to learn how to classify autistic brains, we set up a system of different machine learning classifiers from the set of *n* features selected for each of the 16 data representations. We evaluated six different classifier types and optimized their hyperparameters in order to establish the most accurate parameter classifier model. Those classifiers, including linear and non-linear ones, are: (i) lsvm; (ii) lr; (iii) rf; (iv) lgbm; (v) neural networks (nn); and (vi) radial-basis function SVM (svm). The different types would test different relations between the two feature classes, with the first two being linear estimators. The cross-validated random search is used for hyper-parameter optimization for each of the six classifiers, with k = 5 folds, optimizing for a balanced accuracy score on each test fold to obtain the best parameters on the hyper-search space. Accordingly, these steps are followed for each classifier, for each dataset using only the selected features: split data into k-folds, train four and test one each round, and record the performance of the classifier for each round on the test subset for each parameter configuration. The classifier with the best performance is selected, together with hyperparameters and its maximum average cross-validated score, as well as the standard deviation over folds. A detailed step-by-step guide to the full implementation of the algorithm is provided in Algorithm 1.
**Algorithm****1** Step-by-step rs-fMRI diagnosis algorithm1:∀ **rs-fMRI BOLD signal data:**2:      1.∀ preprocessing strategies:3:           (i) with/without global signal correction.4:           (ii) with/without band-pass filtering.5:      2. Use ∀ atlas ∈ {AAL, TT}, ∀ preprocessing strategy:6:           (i) AAL atlas with 116 brain regions7:           (ii) TT atlas with 97 brain regions8:      3. Calculate the two feature representations for each atlas/strategy:9:            (i) Static functional connectivity matrix FC10:          (ii) Dynamic functional connectivity dFC11:               (I) Use a Gaussian smoothed sliding window over each pair of brain regions to calculate pair-wise dynamic correlations.12:               (II) ∀ pair of brain regions, calculate the fraction of no correlation as $n_{wk}$, and the fraction of strong correlation as $n_{st}$.13:               (III) Use these aggregations for each region pair as the new dynamic functional connectivity feature to create the feature matrix dFC.14:Feature Selection:15:      For each feature representation, run a univariate selector to reduce feature space16:      For each of the four RFE-CV kernels, find the *n* features that provide the highest cross-validated balanced accuracy to be used for each of the kernels.17:Classification:18:      ∀ classifier, for each configuration of hyper-parameters, for each reduced feature representation:19:       (i) Split reduced $X_{select}$, with *n* selected features, into k folds, along with labels *y*.20:       (ii) Calculate the cross-validated score for each hyper-parameters’ configuration.21:      (iii) Determine the best hyper-parameters configuration in terms of score for each classifier.22:      (iv) Output the best classifier/parameters, along with its used *n* features.

### 2.7. Performance Metrics

To measure the performance of machine learning components, different metrics are used to evaluate autism diagnosis, especially when classes are imbalanced on the original dataset or on the output labels. Let TP indicate true positive, TN indicate true negative, FN denote false negative, and FP denote false positive. The following performance metrics are used in this work and defined as follows:Specificity: $\frac{TN}{FP+TN}$Sensitivity (recall): $\frac{TP}{TP+FN}$Accuracy: $\frac{TP+TN}{TP+TN+FP+FN}$Balanced accuracy: average of true positive rate (sensitivity) and true negative rate (specificity).

## 3. Results

Running the explained experimental pipeline on a cluster of 16 high-performance computers, we were able to collect detailed results. Appendix A provides 16 sub-tables of cross-validated test scores, one table for each feature_representation/atlas/strategy, with mean ± standard deviation across folds for each best hyper-parameters for each feature selection kernel (row) and machine learning classifier (column). Here, we highlight what was the best choice for each stage, in addition to the power of this choice, and the best accuracy achieved accordingly.

### 3.1. Significance of Data Representation

In order to test the significance of the configuration of choice, we logged all model scores for each of the five test stages, labeled as (1) feat: (two) FC or dFC, (2) atls: (two) TT or AAL, (3) preprocessing strategy: (four) filt_global, etc., (4) feature selection kernel of the RFE-CV: (four), and (5) machine learning classifier (six). Thus, the labeled table contains up to 1920 results. Three-factor ANOVA, with the calculation of the sum of squares (SS) for the factors, is used to test the effect of the three major factors pre-machine learning: (1) feature representation, (2) atlas, and (3) preprocessing. Full interactions are tested (Table 2), before removing non-significant interactions of *p* > 0.001 and rerunning the test (Table 3). The sum of squares that each factor accounted for did not change greatly due to the nature of the type III sum of squares. In the following part, the effect of each factor is discussed. It is important to note that all configurations (for feature selection, classifier, *…* etc.) are considered here, not only high-performing ones, which would be seen as having lower mean accuracy. For each factor, we aim here to study the effect of it comparatively, not just quantify its performance, especially as the experiments are paired: less-performing configurations would appear in both groups.

#### 3.1.1. Preprocessing Pipeline

The choice of the preprocessing strategy (whether to perform global signal correction, and whether to apply band-pass filtering) has a significant effect on the classification performance ($p≃10^{-6}$). From Table 3, with F=10, we can see that the effect of this factor, although significant, is less important than the other factors. While the difference between the overall mean accuracy for (filt_noglobal, nofilt_global, nofilt_noglobal) is less than 0.2%, the mean filt_global accuracy is higher than 2.5%. See Appendix A for the summarized results across strategy selection. Another important aspect when analyzing the full results listed in Appendix A is that although the max performance is not accompanied by the filt_global strategy, the scores are slightly higher across other models of the strategy.

#### 3.1.2. Atlas Use

In the second level of importance comes the used atlas in brain parcellation (TT or AAL), with F=25.5. The choice of the atlas also has a significant effect on the classification performance ($p≃4\times 10^{-7}$) as we can see in the ANOVA tables. The mean average performance is 2% higher in favor of the AAL atlas. This gives an important indication that the more granular the atlas is (116 areas in comparison to 97 areas), the more informative the features we have (the mean BOLD signal), which leads to better accuracy results. See Appendix A for the summarized results across atlas use.

#### 3.1.3. Dynamic Connectivity

As clearly seen in Table 2 and Table 3, the proposed new feature representation has the biggest effect on the target (accuracy score). This is demonstrated with a high f_value of F≃160 and a highly significant probability of $p≃2\times 10^{-35}$. On average, the dynamic functional connectivity representation we proposed scored an accuracy 5% higher than the conventional functional connectivity. The results represent a key finding of the body of the presented comprehensive experiments in this work.

### 3.2. Model Results

In Appendix A, the file contains four tabs, each for an atlas/feature representation combination, in addition to a summary tab. We can see a trend in performance, with (dFC/AAL) being the best performing in general, then (dFC/TT), (FC/AAL), and (FC/TT) having the least scores. In terms of average scores, as shown in the Appendix A, lsvm kernel of the feature selection is the best performing, and rf is the worst. Both lsvm and lrclassifiers perform the best on average, while rf and lgbm scores come at the end. The top achieved balanced accuracy is 98.8% for lsvm feature selection, lsvm classifier, nofilt_global preprocessing, AAL atlas, and dFC feature representation. This model configuration (preprocessing, feature representation, atlas, feature selection, classifier, and found best parameters) is elected as the best classifier and is investigated more in the following subsections. The original dFC input is composed of 13,340 features, from which the lsvm-based feature selection elects 840 features, which are used in further processing. The best model is cloned, and five-fold cross-validation using a new random split is made to ensure that the results are not specific to the previous k-fold run. Table 4 shows the results of the best configuration model in terms of the accuracy, sensitivity, specificity, and balanced accuracy ± standard deviations.

#### Identified Brain Areas

Appendix A provides the selected features that lead to the best model performance. An especially interesting finding is that identifying the weak/under-connectivity between brain areas is far more important than the very strong correlations, with a percentage of wk features = 81.9%, and strong correlations contributing only to the remaining 18.1% of the selected features. We highlight the first 10 used features, as well as the top frequent brain regions in this selected list in Table 5, while the full list is available in the Appendix A.

## 4. Discussion

We found a striking overlap between the top brain regions associated with ASD in our study with those identified in other fMRI studies. For example, left sup temporal (Temporal_Sup_L) was identified among regions of great contribution to ASD classification [40]. Boddaert et al. [41] also reported superior temporal cortex activation in autistic patients. Several regions of the cingulum have been associated with social cognition and language [42], which is a common deficiency in autistic subjects. Effective Connectivity of areas Postcentral_L, Frontal_Sup_Medial_R, Precuneus_R, SupraMarginal_R, Supp_Motor_Area_L, and Hippocampus_R, were identified as playing an important role in distinguishing between individuals with autism and controls [43]. Regions Temporal_Sup_L, Postcentral_R, Frontal_Sup_Medial_R, Precuneus_R, Cingulum_Ant_R, and Hippocampus_R were among ASD-related regions highlighted by Chen et al. [44]. Children with ASD also showed decreased functional connectivity in the right angular gyrus [45] as part of the cognition network. The same overlap with the selected brain regions can be seen in many other studies [46,47,48,49]. While this work only identifies such disrupted connectivity, an explanation of the impaired development of underlying connectivity can be linked to the same etiology of autism. Spotting those differences in brain connectivity is an important step toward a better understanding of autism.

A comparison of the experimental results of some other existing classification systems in the last five years can be found in Table 6. Our results are clearly higher than most of the literature for the ABIDE-I dataset, taking into consideration that some are only on part of the dataset (not all sites). The added value of this work not only establishes better accuracy on a large dataset but also shows comprehensive experimentation of different blocks along the pipeline. Moreover, the ‘average’ performance across our novel dynamic connectivity feature dFC across all configurations, including other atlas and inferior strategies, is still greater than most of the literature (78.09% average accuracy, see Appendix A). This is a powerful indication of the potential of our proposed work.

We would emphasize, however, that the promising results were obtained and validated on one dataset only, and adding more data should ensure the generalizability of our proposed framework, an intriguing avenue for future research. In conjunction with fMRI, incorporating other modalities such as sMRI or DTI is believed to provide more accurate and specific results, thus improving diagnostic sensitivity, accuracy, and specificity.

## 5. Conclusions

In this paper, a pipelined framework for ASD diagnosis is investigated, based on the analysis of brain rs-fMRI, incorporating the advantages of new dynamic feature representation and machine learning prospects. As a result of the framework presented, a number of objectives were achieved. In addition to providing an accurate state-of-the-art diagnosis of ASD based on a public dataset, it can also be used for identifying the brain regions contributing to such diagnosis. In conjunction with the framework’s diagnosis, these spotted brain areas can be reported to the physician early on so that he or she may make a more informed decision in the future. Having a better understanding of the brain abnormalities associated with autism is one of the key aspects of this study. Moreover, the comprehensive experiments made in this work pave the way for the next researchers to understand the effect of different choices along the neuroimaging machine learning pipeline on diagnostic accuracy, filling an important gap in knowledge that would help to speed the development of such CAD solutions. A further advantage of the system is that it easily scales up: more subjects can be preprocessed and their features calculated independently, and the addition of an additional mode of imaging over the same subject, such as structural MRI, can be incorporated into the feature selection stages.

It is anticipated that further development of MRI imaging-based AI approaches will enhance subjectivity in the clinical information available for ASD diagnosis in the future, enabling more accurate and expedient diagnoses to be made, enhancing our understanding of the underlying diagnoses of autism and moving us towards an automated system for an autism assessment.

## Figures and Tables

**Figure 1 bioengineering-10-00056-f001:**
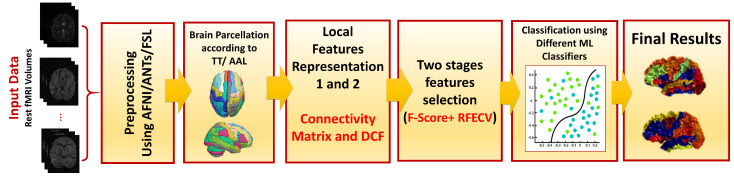
Illustration of the full steps of the adopted framework, including preprocessing, brain parcellation, feature representation, feature selection, and machine learning (ML) classification.

**Figure 2 bioengineering-10-00056-f002:**
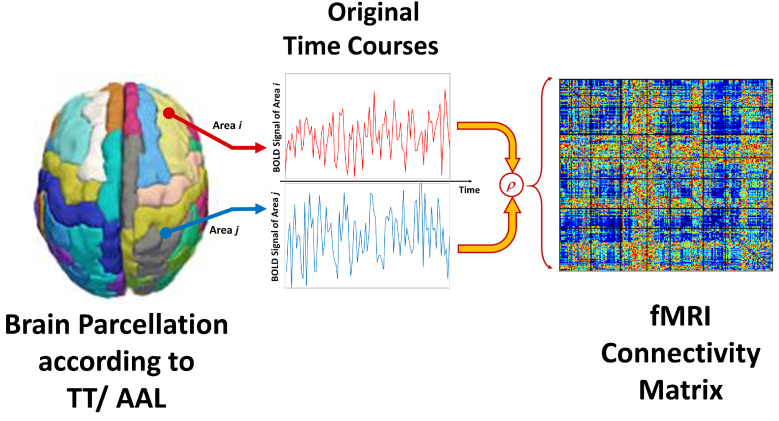
Visual diagram of the calculations of the static functional connectivity FC representation, extracted from two different brain areas.

**Figure 3 bioengineering-10-00056-f003:**
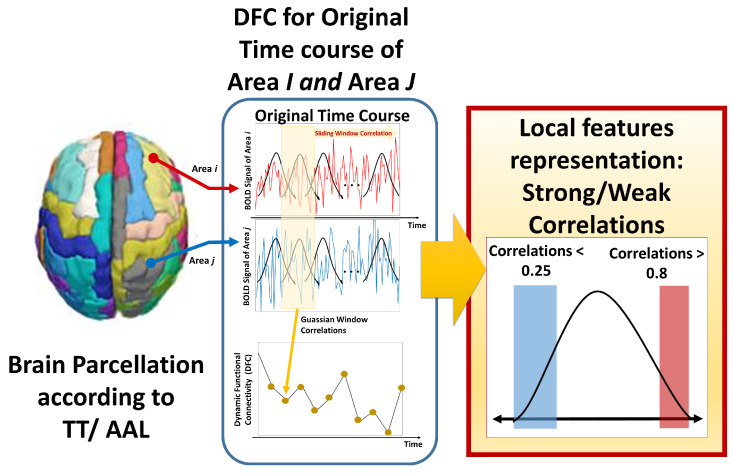
Visual diagram of the calculations of the dynamic functional connectivity dFC representation, extracted from two different brain areas.

**Table 1 bioengineering-10-00056-t001:** Demographics summary of the 884 subjects used in this study, showing sex counts (M: male, F: female), age at scan in years, and full-scale IQ (FIQ).

	ASD Group (*n* = 408)		TD Group (*n* = 476)	
	M = 358, F = 50		M = 388, F = 88	
	**AGE**	**FIQ**	**AGE**	**FIQ**
count	408	379	476	442
mean	17.69	106.19	16.79	111.28
std	8.93	17.01	7.35	12.48
min	7	41	6.47	73
max	64	148	56.2	146

**Table 2 bioengineering-10-00056-t002:** Multifactorial (3-way) ANOVA results. Feat denotes feature representation, Atls denotes the used atlas, and Strat is one of the four preprocessing strategies. sum_sq, df, F, PR are standard result names of sum of squares, degree of freedom, F_score, and *p*_value of this F_score, for each factor.

	sum_sq	df	F	PR (>F)
C(Feat, Sum)	0.941852	1.0	163.004715	$7.364232\times 10^{-36}$
C(Strat, Sum)	0.177957	3.0	10.266230	$1.053597\times 10^{-06}$
C(Atls, Sum)	0.149164	1.0	25.815521	$4.129224\times 10^{-07}$
C(Feat, Sum):C(Strat, Sum)	0.117433	3.0	6.774646	$1.528091\times 10^{-04}$
C(Feat, Sum):C(Atls, Sum)	0.013784	1.0	2.385576	$1.226282\times 10^{-01}$
C(Strat, Sum):C(Atls, Sum)	0.034962	3.0	2.016924	$1.095533\times 10^{-01}$
C(Feat, Sum):C(Strat, Sum):C(Atls, Sum)	0.084563	3.0	4.878417	$2.210611\times 10^{-03}$

**Table 3 bioengineering-10-00056-t003:** Multifactorial (3-way) ANOVA test results after removing insignificant interactions.

	sum_sq	df	F	PR (>F)
C(Feat, Sum)	0.936316	1.0	160.682470	$2.134089\times 10^{-35}$
C(Filt, Sum)	0.179172	3.0	10.249322	$1.078933\times 10^{-06}$
C(Atls, Sum)	0.148105	1.0	25.416432	$5.062110\times 10^{-07}$
C(Feat, Sum):C(Filt, Sum)	0.117208	3.0	6.704737	$1.686921\times 10^{-04}$

**Table 4 bioengineering-10-00056-t004:** Cross-validated test results of the best configuration model.

Metric	Accuracy	Sensitivity	Specificity	Balanced Accuracy
5-fold value	0.988 ± 0.004	0.987 ± 0.008	0.989 ± 0.007	0.988 ± 0.004

**Table 5 bioengineering-10-00056-t005:** Summary of first 10 selected features as well as the top frequent brain regions in this list.

Index	First Selected Features	Top Frequent Region Names	Frequency
1	Precentral_L__Rolandic_Oper_R_wk	Temporal_Sup_L	24
2	Precentral_L__Supp_Motor_Area_L_st	Postcentral_R	23
3	Precentral_L__Fusiform_L_wk	Frontal_Sup_Medial_R	22
4	Precentral_L__Parietal_Inf_L_wk	Precuneus_R	22
5	Precentral_L__SupraMarginal_R_st	SupraMarginal_R	21
6	Precentral_L__Precuneus_R_wk	Frontal_Sup_R	21
7	Precentral_L__Temporal_Sup_L_wk	Cingulum_Ant_R	21
8	Precentral_L__Temporal_Pole_Sup_R_wk	Supp_Motor_Area_L	21
9	Precentral_L__Cerebelum_Crus2_R_wk	Angular_R	21
10	Precentral_R__Frontal_Inf_Tri_L_wk	Hippocampus_R	21

**Table 6 bioengineering-10-00056-t006:** Comparison of results of some existing ASD classification methods using the ABIDE-I dataset.

Article	Used Classifier	Achieved Accuracy
Abraham et al., 2017 [50]	SVM	67.0%
Guo et al., 2017 [51]	Deep neural networks with feature selection (DNN-FS)	86.4%
Kam et al., 2017 [52]	Discriminative restricted Boltzmann machines (DRBM)	80.8%
Sadeghi et al., 2017 [53]	SVM	92%
Spera et al., 2019 [54]	SVM	71.0%
Tang et al., 2019 [55]	SVM	62.6%
Wang et al., 2020 [56]	MLP and a voting strategy	74.5%
Rakić et al., 2020 [49]	Ensemble of classifiers	85.0%
Subah et al., 2021 [57]	DNN	87.0%
Al-Hiyali et al., 2021 [58]	SVM, K-nearest neighbors (KNN)	85.9%
Yin et al., 2021 [59]	Autoencoders, CNN, DNN	79.2%
Chu et al., 2022 [60]	Multi-scale graph convolutional network (GCN)	79.5%
Yang et al., 2022 [61]	LR, SVM, DNN, supervised learning classifier	69.4%
Ding et al., 2022 [62]	low-rank domain adaptive method with inter-class difference constraint	75.5%
Proposed Work	Best: LSVM	98.8%

## Data Availability

This work uses rs-fMRI data from the ABIDE-I preprocessed dataset, publicly available at http://fcon_1000.projects.nitrc.org/indi/abide/abide_I.html, accessed on 1 July 2022. The website also contains demographic information and scanning parameters that were used.

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
