# Peer review of "Understanding the Role of Connectivity Dynamics of Resting-State Functional MRI in the Diagnosis of Autism Spectrum Disorder: A Comprehensive Study"

_bioengineering, 2023, doi:10.3390/bioengineering10010056_

Round 1

Reviewer 1 Report

This study explores neuroimaging research for ASD by providing evidence for widespread functional and structural abnormalities in the complex neurodevelopmental disorder.

There has been tremendous utilization of this technique since the first publications using rs-fMRI in ASD research in 2006. Although this upsurge of interest in ASD research is positive, it comes with the formidable challenge of explaining contradictory reports of ASD connectivity. What is the research gap?

In the Discussion, please explain whether altered connectivity is a result of a failure to generate connections, inappropriate pruning, or creation of new connections?

What are the limitations of the study?

Conclusions to be written in a separate section.

Author Response

We would like to thank the reviewer for his valuable comment and positive feedback and constructive comments. Here is a detailed response to the raised points:

Comment 1: There has been tremendous utilization of this technique since the first publications using rs-fMRI in ASD research in 2006. Although this upsurge of interest in ASD research is positive, it comes with the formidable challenge of explaining contradictory reports of ASD connectivity. What is the research gap?

Author’s Response: Thank you for the valuable comment and positive feedback. This work addresses using resting state functional brain imaging features for the diagnosis of Autism. It allows not only an accurate diagnosis but also the identification of the brain regions contributing to this decision. More importantly, it quantifies the effect of different choices along the neuroimaging machine learning pipeline on diagnostic accuracy, including the use of dynamic over static connectivity, which is important for future research. In the revised manuscript, a paragraph has been added in the Introduction section, lines 104-115, page 3, to address the research gap, the limitations of the current work, and how to fill this gap. In addition, the conclusion is updated to better clarify this part, lines 378-382, page 11. 

Comment 2: In the Discussion, please explain whether altered connectivity is a result of a failure to generate connections, inappropriate pruning, or creation of new connections?

Author’s Response: Thank you for the valuable comment. While this work only identifies such disrupted connectivity, an explanation of the impaired development of underlying connectivity can be linked to the same etiology of autism. In the revised manuscript, we added an illustration in the discussion part to tackle this point. Please see lines 351-354, page 10, in the Discussion section.

Comment 3: What are the limitations of the study?

Author’s Response: Thank you for the valuable comment. The main limitation of the proposed method is that it was tested on only one dataset. There is a need to test it on other datasets to ensure the generalizability of the method. In addition, the method is limited to the features that are extracted from sfMRI. Integrating features using multi-modal MRI images would be more accurate. In the revised manuscript, the discussion section has been updated to include the limitations of the proposed work, please, check the last paragraph in the Conclusions section, lines 364-368, page 10.

Comment 4: Conclusions to be written in a separate section

Author’s Response: Thank you for the valuable comment. We updated the manuscript by adding a separate Conclusion Section and updated the discussions accordingly. Please see Sections 4 and 5 on pages 10 and 11, respectively.

Reviewer 2 Report

Review: Summary of the Study:

This study by ElNakieb et al. is a retrospective study evaluating the existing data retrieved from the Autism Brain Imaging Data Exchange of resting state fMRI (rsfMRI) evaluating and understanding to correlate the brain imaging using engineering tools to interpret the deviance of the atypical brain image using a novel dynamic connectivity algorithms, using stepwise regressors, in association with the subjective assessment, such as Autism Diagnostic Observation Schedule and Revised methods to better assist physicians to make an informed decision. The method the authors provided in this study gives the confidence to achieve diagnostic accuracy of about 78%.

Comments to the authors:

This is a retrospective analysis of the existing data and it provides a comprehensive attempt to reanalyze rsfMRI from the data extracted from the database. The description of the analysis is well explained and a significant estimation of 78% accuracy was reported by using their analyses.

Some of the concerns the authors are encouraged to answer are:

1.     There is nowhere a mention of how many rsfMRI or sMRI were included in this screening analysis. While the authors provide the database where they extracted the data, in the analyses where they were able to distinguish between atypical and typical brain images if so, how many rsfMRI of each group were included in the analysis?

2.     When such a large set of data are analyzed and relevant to medical care for usability, it’d be necessary to include the demographics of the subject included in the analysis either in the form of a table or a pie diagram, or a histogram of the subjects used for this analysis even though the brain images vary due to the spectrum. This would provide a better understanding of how such analysis can be interpreted to address specific needs for diagnostic criteria.

3.     As stated in Figure 1, include another panel of the subject (rsfMRI) selection criteria. The subjects' brain images extracted from the database needs to be detailed. 

4.     The figures provide a title, it would be necessary to provide a figure legend, a description of what the readers are viewing and if the panels are represented in color, the description of the color would benefit readers to have a better understanding of what is presented in each panel/figure. 

5.     In table 1, please describe how were the insignificant interactions removed. 

6.     It is nice that the authors made a mention that the lower-density signals were also considered. So, in density network analysis, have they identified any false positives in the regions with higher density signals?

7.     In Table 5, the authors can provide their achieved accuracy at the bottom since the references are arranged chronologically.

8.     Parenthesis with reference number should not replace a word. In other words, it can only be cited, consider revising the following sentences:

Ln48: diagnosing schizophrenia using the approaches described in [14]. – revise

Ln58: There is evidence in [17], and [19] for the hypothesis of underconnectivity. - revise

Ln60: In another study by [20], - provide the name of the author or revise the sentence to give the reference in parenthesis at the end of the sentence.

Ln62: In [21], abnormalities were reported 

Ln69: Furthermore, the presence of altered 69 connectivity was verified in [23]. 

Ln71: Furthermore, a study by [24] identified 71 hyperconnectivity 

Ln178: Following the literature, [34], a Gaussian

Ln312: For example, in [40], left

Ln314: Researchers in [41] also reported

9.     It is preferred to separate the Discussion and Conclusion rather than combine them. The Conclusion would provide a comprehensive overview of your experience although the results obtained, are elaborated in the Discussion, the conclusion will precisely convey the progress made, the limitations, and the future goal hence, the suggestion is to separate the Discussion and Conclusion for better readability and comprehension.

Author Response

We would like to thank the reviewer for his valuable comment and positive feedback and constructive comments. Here is a detailed response to the raised points:

Comment 1: There is nowhere a mention of how many rsfMRI or sMRI were included in this screening analysis. While the authors provide the database where they extracted the data, in the analyses where they were able to distinguish between atypical and typical brain images if so, how many rsfMRI of each group were included in the analysis?

Response: Thank you for your comments. Only rs-fMRI was used for each of the 884 subjects that were included. All of the 884 rs-fMRI data were used. The ASD group contains 408 subjects, whereas the TD group contains 476 subjects. In the revised manuscript, we added a subsection (2.1. Dataset, pages 3 and 4, lines 131-142) to address the raised concern. In addition, Table 1 on page 4 is added in the revised manuscript to include the number of subjects in each group as well as their demographics.

Comment 2: It’d be necessary to include the demographics of the subject included in the analysis either in the form of a table or a pie diagram, or a histogram of the subjects. 

Response: Thank you for the valuable comment. We updated the Materials and methods section by adding a summary of the demographics. Please see subsection 2.1. Dataset, Table 1 on page 4.

Comment 3: As stated in Figure 1, include another panel of the subject (rsfMRI) selection criteria.

Response: There were no subject exclusion criteria. All of the 884 available rs-fMRI subjects collected through the different ABIDE-I sites were used. Selection and exclusions were part of the original dataset collection, not a step of this proposed framework. The revised manuscript has clarified this point in subsection 2.1, lines 135-136, page 4.

Comment 4: The figures provide a title, it would be necessary to provide a figure legend, a description of what the readers are viewing and if the panels are represented in color, the description of the color would benefit readers to have a better understanding of what is presented in each panel/figure.

Author’s Response: Thank you for the valuable comment. The different colors in the different plots inside the figures were only for visualization purposes, and don’t carry any extra information, hence no legends were used. To make the figures more readable and address the reviewer's comment, we updated the figure captions to be more descriptive and allow a better understanding of what is presented. Please, check the updated figures 1, 2, and 3 captions, on pages 5 and 6.

Comment 5: In table 1, please describe how were the insignificant interactions removed.

Response: Thank you for this comment. Only significant interactions, with strict p-value of p<0.001, were used for the second step, on which we re-ran the multifactorial ANOVA with only those significant interactions, which is the typical procedure. In the revised manuscript, we updated this part to describe more clearly how the insignificant interactions were removed, subsection 3.1, page 8, lines 277-278.

Comment 6: It is nice that the authors made a mention that the lower-density signals were also considered. So, in density network analysis, have they identified any false positives in the regions with higher density signals? 

Response: 

Thanks a lot for this valuable comment. The overall goal of this study is to provide a diagnostic framework and find the distinctive dynamic connectivity features. In this regard, we found out that the weak/ under-connectivity between brain areas was more important than the very strong correlations. It will be very interesting to include a graph-theoretical approach for density network analysis in our future work. Please, check subsection 3.2.1, page 10, lines 331,332 in the revised manuscript. 

Comment 7: In Table 5, the authors can provide their achieved accuracy at the bottom since the references are arranged chronologically?

Response: Thank you for your comment. We added our achieved accuracy at the bottom of the table. Please, check Table 6, page 11 on the revised manuscript. 

Comment 8: Parenthesis with reference number should not replace a word. 

Response:  Thank you for the valuable comment. We updated the citations to account for this feedback throughout the manuscript, and are highlighted in yellow, lines 44, 53, 56, 57, 66, 68, 191, and 233. 

Comment 9: It is preferred to separate the Discussion and Conclusion rather than combine them.

Response: Thank you for the valuable comment. We updated the manuscript by adding a separate conclusion section and updated the discussions accordingly. Please see sections  4, and 5  on pages 10 and 11, respectively. 

Reviewer 3 Report

  1. What is the need of this research, justify please 
  2. Explain the research model in depth
  3. Pseudo code addition will improve better understanding the authors work
  4. Further details on experiments , results required.
  5. English must be double-checked with native speaker
  6. Include confusion matrix please 
  7. What are the performance metrics ?
  8. Which dataset used ? include one sub-section on its details.
  9. How your proposed model results are better in state of art, compare results on same dataset. 

Author Response

The authors would like to thank the reviewer for the valuable feedback. Here is a detailed response to the raised points:

Comment 1: What is the need of this research, justify please.

Response: Thank you for the valuable comment and positive feedback. This work addresses using resting state functional brain imaging features for the diagnosis of Autism. It allows not only an accurate diagnosis but also the identification of the brain regions contributing to this decision. More importantly, it quantifies the effect of different choices along the neuroimaging machine learning pipeline on diagnostic accuracy, including the use of dynamic over static connectivity, which is important for future research. In the revised manuscript, a paragraph has been added in the Introduction section, lines 104-115, page 3, to address the research gap, the limitations of the current work, and how to fill this gap. In addition, the conclusion is updated to better clarify this part, lines 378-382, page 11. 

Comment 2: Explain the research model in depth.

Response: Thank you for the valuable comment. We updated the materials and methods to include an extra subsection ‘2.2 Proposed Framework, page 4, lines 143-148 that present our adopted research model. 

Comment 3: Pseudo code addition will improve better understanding the authors work.

Response: Thank you for the valuable comment. We updated the materials and methods section to also include a pseudo-code. Please see added Algorithm 1, page 7.

Comment 4: Further details on experiments, results required.

Response: Thank you for the valuable comment. To fully explain the experiments and results, the revised manuscript has been intensively updated. Multiple subsections (2.1 (lines 131-142, pages 3 and 4), 2.2 (143-148, page 4), and 2.7 (251-261, pages 7-8) have been added to the methods section to include the details of the number of subjects and their demographics for both autism and normal groups, the details of the proposed framework, and the details of the performance metrics. The overall algorithm has been added (Algorithm 1, page 7). In addition, the result section has been updated to include more details on the ANOVA test (page 8, lines 275-278). The Discussion and Conclusions sections have been separated in the revised manuscript. The updated Discussion section (page 10 and 11) highlight the results/ findings introduced in this work, including more details about the results, their discussions, and the limitations of the work. 

Comment 5: English must be double-checked with native speaker.

Response: Thank you for the valuable comment. We have thoroughly revived the manuscript after the updates with a native speaker. All corrections are highlighted in yellow, in the revised manuscript.

Comment 6: Include confusion matrix please. 

Response: Thanks a lot for this valuable comment. Since our results, are cross-validated over 5-folds, it is not common to include 5 different confusion matrices for each model. More commonly, sensitivity (TP/[TP+FN]), specificity (TN/[TN+FP]), accuracy ([TP+TN]/all), and balanced accuracy (average of sensitivity and specificity) are used, which give a complete picture of the weights of different classes, where a standard deviation for each metric is added to summarize the effect of using different testing folds and indicate any biasing towards the selection of training set. The complete set of performance metrics with their standard deviations have been evaluated and highlighted in the revised manuscript, Table 4, page 9.

Comment 7: What are the performance metrics ?

Response: Thank you for your feedback. We added a subsection (2.7: Performance metrics, pages 7-8, lines 251-261) to introduce all of the used performance metrics. 

Comment 8: Which dataset used? include one sub-section on its details. 

Response:  Thank you for the valuable comment. All rs-fMRI subjects from the ABIDE-I dataset are used. We added a subsection (2.1. Dataset, pages 3 and 4, lines 131-142) to address the details of the data and its demographics.

Comment 9: How your proposed model results are better in state of art, compare results on same dataset.

Response: Thank you for the valuable comment. We updated the manuscript with Table 6, page 11, which contains a comparison results, with our results and some previous work in the same dataset (ABIDE).

Reviewer 4 Report

The paper written by the following Authors: Yaser ElNakieb, Mohamed T. Ali, Ahmed Elnakib, Ahmed Shalaby, Ali Mahmoud, Ahmed Soliman, Gregory Neal Barnes, and Ayman El-Baz, entitled “Understanding the Role of Connectivity Dynamics of Resting-State Functional MRI in the Diagnosis of Autism Spectrum Disorder: A Comprehensive Study” presents an interesting study on neuroimaging in the diagnosis of autism.

Although the paper is interesting, I have some major concerns:

Abstract

The abstract is lacking the aim of the study, material and methods description as well as an informative conclusion. It should be written in more details.

Material and Methods

- The parameters of applied MRI scans should be included in the manuscript.

- Which brain regions were analyzed?

Results

- Authors mentioned “Three-factor ANOVA with the calculation of the sum of 250 squares (SS) for the factors, was used to test the effect of the major three factors pre-machine 251 learning: feature representation, atlas, and preprocessing.”. Which factors were tested?

- Application of Atlas should be explained in more details. Is there any extrapolation of presented results?

Discussion

Limitation to the study should be added at the end of the discussion part.

Author Response

The authors would like to thank the reviewer for the valuable feedback. Here is a detailed response to the raised points:

Comment 1: The abstract is lacking the aim of the study, material, and methods description as well as an informative conclusion. It should be written in more details.

Response: Thank you for the valuable comment. We updated the abstract adding a concise aim, description, and conclusion (Page 1, lines 1-17).  

Comment 2: The parameters of applied MRI scans should be included in the manuscript.

Response: Thank you for the valuable comment. Since the full list of parameters is long [1-2 pages per site], with many differences between sites [17 different sites], we included the publicly available link in the text in subsection 2.1, lines 140-141.

Comment 3: Which brain regions were analyzed?

Response: Thank you for the valuable comment. All brain regions defined by each of the two used brain atlases were analyzed. The revised manuscript has been updated to address this issue, subsection 2.3, page 4, lines 167-170.

Comment 4: Authors mentioned “Three-factor ANOVA with the calculation of the sum of squares (SS) for the factors, was used to test the effect of the major three factors pre-machine learning: feature representation, atlas, and preprocessing.”. Which factors were tested?

Response: Thanks a lot for this valuable comment. The three factors tested are: 1) feature representation, 2) atlas, and 3) preprocessing, as stated. Their effect on the resulting accuracy, as well as interactions, is tested. Full interactions were first tested, then non-significant interactions below 0.001 have been removed. The revised manuscript has been updated to address this issue, subsection 3.1, page 8, lines 275-278.

Comment 5: Application of Atlas should be explained in more details. Is there any extrapolation of presented results.

Response: The use of the atlas in our work, as illustrated in the subsection "2.3 preprocessing", is to localize the average BOLD signal to different brain regions. We added more explanation to the use of the atlas, highlighted in yellow (subsection 2.3, lines 165-166 and lines 169-170, page 4). Currently, there is no extrapolation in the results presented.

Comment 6: Limitation to the study should be added at the end of the discussion part.

Response: Thank you for the valuable comment. The main limitation of the proposed method is that it was tested on only one dataset. There is a need to test it on other datasets to ensure the generalizability of the method. In addition, the method is limited to the features that are extracted from sfMRI. Integrating features using multi-modal MRI images would be more accurate. In the revised manuscript, the discussion section has been updated to include the limitations of the proposed work, please, check the last paragraph in the Conclusions section, lines 364-368, page 10.

Round 2

Reviewer 3 Report

The authors have improved some parts of the article but did not compare the results with the latest published techniques. I suggest authors to compare results with the following techniques.

Yousaf, K., Mehmood, Z., Awan, I. A., Saba, T., Alharbey, R., Qadah, T., & Alrige, M. A. (2020). A comprehensive study of mobile-health based assistive technology for the healthcare of dementia and Alzheimer’s disease (AD). Health Care Management Science23(2), 287-309.

Ramzan, F., Khan, M. U. G., Rehmat, A., Iqbal, S., Saba, T., Rehman, A., & Mehmood, Z. (2020). A deep learning approach for automated diagnosis and multi-class classification of Alzheimer’s disease stages using resting-state fMRI and residual neural networks. Journal of medical systems44(2), 1-16.

Ahmad, M. F., Akbar, S., Hassan, S. A. E., Rehman, A., & Ayesha, N. (2021, November). Deep Learning Approach to Diagnose Alzheimer’s Disease through Magnetic Resonance Images. In 2021 International Conference on Innovative Computing (ICIC) (pp. 1-6). IEEE.

Sajjad, M., Ramzan, F., Khan, M. U. G., Rehman, A., Kolivand, M., Fati, S. M., & Bahaj, S. A. (2021). Deep convolutional generative adversarial network for Alzheimer's disease classification using positron emission tomography (PET) and synthetic data augmentation. Microscopy Research and Technique84(12), 3023-3034.

Author Response

We appreciate the time and effort that the reviewer has dedicated to providing his valuable feedback on our manuscript. We have done changes to reflect the extra suggestions provided by the reviewer. These changes have been highlighted in yellow in the manuscript.

Comment 1: There has been tremendous utilization of this technique since the first publications using rs-fMRI in ASD research in 2006.

The authors have improved some parts of the article but did not compare the results with the latest published techniques. I suggest authors to compare results with the following techniques.

  • Yousaf, K., Mehmood, Z., Awan, I. A., Saba, T., Alharbey, R., Qadah, T., & Alrige, M. A. (2020). A comprehensive study of mobile-health based assistive technology for the healthcare of dementia and Alzheimer’s disease (AD). Health Care Management Science, 23(2), 287-309.
  • Ramzan, F., Khan, M. U. G., Rehmat, A., Iqbal, S., Saba, T., Rehman, A., & Mehmood, Z. (2020). A deep learning approach for automated diagnosis and multi-class classification of Alzheimer’s disease stages using resting-state fMRI and residual neural networks. Journal of medical systems, 44(2), 1-16.
  • Ahmad, M. F., Akbar, S., Hassan, S. A. E., Rehman, A., & Ayesha, N. (2021, November). Deep Learning Approach to Diagnose Alzheimer’s Disease through Magnetic Resonance Images. In 2021 International Conference on Innovative Computing (ICIC) (pp. 1-6). IEEE.
  • Sajjad, M., Ramzan, F., Khan, M. U. G., Rehman, A., Kolivand, M., Fati, S. M., & Bahaj, S. A. (2021). Deep convolutional generative adversarial network for Alzheimer's disease classification using positron emission tomography (PET) and synthetic data augmentation. Microscopy Research and Technique, 84(12), 3023-3034.

Authors’ Response: Thank you for this valuable comment. While all the four recommended papers are focus on Alzheimer’s disease, our work focuses on ASD (Autism) diagnosis using rs-fMRI. Due to the importance of the suggested references, we cited the papers that use the same modality (rs-fMRI) as our work, in the literature review (Introduction Section, lines 44 through 45, page 2). In addition, to follow up with the reviewer suggestion, we updated Table 6 (page 11) to include more recent references from 2022 on ASD on the same dataset. The complete comparison to different rs-fMRI based ASD work on the same dataset from 2017-2022 can be found on the discussion section (Table 6, page 11).

Reviewer 4 Report

I accept the manuscript in present form.

Author Response

Comment 1: I accept the manuscript in present form.

Authors’ Response: The authors would like to thank the reviewer for his positive and valuable feedback.